# Mechanisms of the Epithelial–Mesenchymal Transition and Tumor Microenvironment in *Helicobacter pylori*-Induced Gastric Cancer

**DOI:** 10.3390/cells9041055

**Published:** 2020-04-23

**Authors:** Jacek Baj, Izabela Korona-Głowniak, Alicja Forma, Amr Maani, Elżbieta Sitarz, Mansur Rahnama-Hezavah, Elżbieta Radzikowska, Piero Portincasa

**Affiliations:** 1Chair and Department of Anatomy, Medical University of Lublin, 20-090 Lublin, Poland; amrmaanni@gmail.com; 2Department of Pharmaceutical Microbiology with Laboratory for Microbiological Diagnostics, Medical University of Lublin, Chodzki 1 Street, 20-093 Lublin, Poland; iza.glowniak@umlub.pl; 3Chair and Department of Forensic Medicine, Medical University of Lublin, 20-090 Lublin, Poland; aforma@onet.pl; 4Chair and 1st Department of Psychiatry, Psychotherapy and Early Intervention, Medical University of Lublin, Gluska Street 1, 20-439 Lublin, Poland; elzbietaa.sitarz@gmail.com; 5Chair and Department of Oral Surgery, Medical University of Lublin, 20-081 Lublin, Poland; mansur.rahnama@umlub.pl; 6Department of Plastic Surgery, Central Clinical Hospital of the MSWiA in Warsaw, 01-211 Warsaw, Poland; elzbieta.radzkowska@gmail.com; 7Clinica Medica A. Murri, Department of Biomedical Sciences and Human Oncology, University of Bari Aldo Moro Medical School, 70126 Bari, Italy; piero.portincasa@uniba.it

**Keywords:** *Helicobacter pylori* infection, EMT, epithelial–mesenchymal transition, gastric cancer, virulence factors

## Abstract

*Helicobacter pylori* (*H. pylori*) is one of the most common human pathogens, affecting half of the world’s population. Approximately 20% of the infected patients develop gastric ulcers or neoplastic changes in the gastric stroma. An infection also leads to the progression of epithelial–mesenchymal transition within gastric tissue, increasing the probability of gastric cancer development. This paper aims to review the role of *H. pylori* and its virulence factors in epithelial–mesenchymal transition associated with malignant transformation within the gastric stroma. The reviewed factors included: *CagA* (cytotoxin-associated gene A) along with induction of cancer stem-cell properties and interaction with YAP (Yes-associated protein pathway), tumor necrosis factor *α*-inducing protein, Lpp20 lipoprotein, Afadin protein, penicillin-binding protein 1A, microRNA-29a-3p, programmed cell death protein 4, lysosomal-associated protein transmembrane 4β, cancer-associated fibroblasts, heparin-binding epidermal growth factor (HB-EGF), matrix metalloproteinase-7 (MMP-7), and cancer stem cells (CSCs). The review summarizes the most recent findings, providing insight into potential molecular targets and new treatment strategies for gastric cancer.

## 1. Introduction

*Helicobacter pylori* (*H. pylori*) is a helix-shaped, Gram-negative, microaerophilic, flagellated bacterium that is capable of biofilm formation and converting from spiral to coccoid form [1,2,3,4,5,6]. It is a highly invasive microorganism responsible for one of the highest prevalences of chronic infections worldwide, even though over 80% of infected patients remain asymptomatic [7,8,9,10,11,12,13]. *H. pylori* pathogenesis is due to several virulence factors including urease, cytotoxin-associated gene A (*CagA*), vacuolating cytotoxin (VacA), outer inflammatory protein A (OipA), duodenal ulcer promoting gene A (*DupA*), neutrophil-activating protein A (NAP), heat shock proteins (Hsp10, Hsp60), and sialic acid-binding adhesin (SabA) [14,15,16,17,18,19,20,21,22,23,24].

*H. pylori* is classified as a “group 1” carcinogen according to the World Health Organization (WHO) [25]. Gastric cancer is one of the most common diseases related to *H. pylori* infection, responsible for approximately 75% of gastric cancer incidents worldwide [26,27,28,29]. The discrepancy in the severity of gastric cancer is associated with many factors and mechanisms, among which various genotypes of *H. pylori* strains play a role [30,31,32,33,34,35,36,37,38,39,40]. *H. pylori* induces other diseases of the alimentary tract, including gastritis, peptic ulcer disease, mucosa-associated lymphoid tissue lymphoma, GERD symptoms, and dyspepsia [41,42,43,44,45,46]. There is an increasing interest in the role of *H. pylori* in the pathogenesis of ischemic heart disease, diabetes mellitus, and Alzheimer’s disease [47,48,49,50].

Epithelial–mesenchymal transition (EMT) is the process of acquisition of the mesenchymal properties by epithelial cells involved in metastasis, invasion, and progression of various cancers (Figure 1) [51].

As a physiological process, EMT is observed during organogenesis, tissue development, remodeling, and wound healing [52,53,54]; contrarily, any deregulations might induce carcinogenesis [55,56]. EMT-induced carcinogenesis is the prevalent cause of various malignancies including head and neck squamous cell carcinoma, papillary thyroid carcinoma, lung, pancreatic, gastric, ovarian, prostate, and breast cancer [57,58,59,60,61,62,63,64,65,66,67,68].

During this process, epithelial cells undergo a series of biochemical changes, which lead to the loss of polarity and migratory capacity of cells, resulting in cell shape changes (cell elongation). EMT promotes the transformation of immobile epithelial cells into motile mesenchymal cells, enhancing the metastatic properties [69,70]. Further, adherens and tight junctions become impaired, resulting in a mesenchymal phenotype [12,71,72,73]. Altered E-and N-cadherin levels and the following β-catenin activation promote the expression of many tumor-associated proteins, including cyclin D1, CD44, or *c-MYC* [54,74,75,76,77,78,79]. A transformation of cell phenotype enhances the migratory properties, invasiveness, and apoptosis resistance of cells [80]. Moreover, EMT is involved in the induction of cancer stem cell properties, which leads to chemoresistance and tumor dormancy [81,82,83].

*H. pylori* infection significantly affects the gastric microenvironment by induction of several inflammatory responses via infiltrating macrophages, neutrophils, regulatory T-cells, and natural killer cells [84,85]. Inflammatory mediators such as cytokines, chemokines, and metalloproteinases that are released by gastric and infiltrating cells promote the EMT process within gastric cells; transforming growth factor β (TGF-β) is probably one of the most relevant EMT inducers [86,87,88]. Thus, chronic inflammation might significantly contribute to EMT progression and carcinogenesis [89,90,91]. A significant number of *H. pylori* virulence factors are considered being associated with the promotion of EMT in gastric cells, which consequently causes neoplasia and malignant transformation. This review summarizes several mechanisms associated with epithelial–mesenchymal transition, gastric tumor microenvironment, and the influence of *H. pylori* infection, although some described mechanisms are not only *H. pylori*-specific. Even though *H. pylori*-induced carcinogenesis is not fully understood, several mechanisms have already been deciphered.

## 2. Cytotoxin-Associated Gene A

### 2.1. CagA and EMT

Among many virulence factors of *H. pylori*, cag pathogenicity island (cagPAI) probably plays a key role in carcinogenesis [92,93,94]. It encodes a type 4 secretion system (T4SS) and the *CagA* oncoprotein [95,96]. The T4SS forms a pilus that allows the injection of *CagA* into a cell, transforming its shape into the so-called “hummingbird phenotype” characterized by an elongated cell shape commonly observed in EMT [97,98]. Injection of the *CagA* into the cell via the T4SS induces signal transduction, with one of the most relevant mecahnisms being the nuclear factor κB (NF-κB) signaling pathway involving extracellular regulated kinases 1/2 (ERK-1/2) [99,100,101]. These kinases are involved in the conformational changes of the cytoskeleton, which might enhance the EMT process [102]. The inhibition of ERK and c-Jun N-terminal kinase (JNK) results in the lower expression of the ‘hummingbird’ phenotype induced by *H. pylori* [103]. *CagA* in host cells is tyrosine phosphorylated and interacts with protein tyrosine phosphatase 2 (SHP-2), also inducing the progression of the “hummingbird phenotype” [104,105]. *CagA* enhances EMT via the stabilization of Snail protein, which is essential in carcinogenesis, mainly by the reduction of glycogen synthase kinase-3 (GSK-3) activity [106]. *CagA*-positive *H. pylori* strains with *CagA* containing phosphorylation-functional EPIYA motifs present significantly higher expression of mesenchymal markers such as vimentin, Snail, and ZEB-1 and the stem cell marker CD44 [96,107,108,109,110]. Many studies have shown that *CagA*-positive *H. pylori* strains induce a higher probability of gastric carcinogenesis and induction of EMT process [111,112,113]. Incidents of *CagA*-positive *H. pylori* infection present poor clinical outcome, and higher invasion and metastatic characteristics [114]. Besides, there is an increasing interest in microRNAs (miRNAs), since these are reported to play a role in gastric carcinogenesis and progression [115,116].

### 2.2. CagA and Cancer Stem Cell Properties

Recent research suggests that cells that undergo EMT obtain the ability to acquire cancer stem cell (CSC) properties [117,118,119,120,121,122,123]. The main source of gastric CSCs includes stem cells and progenitor cells; other studies suggest that CSCs originate from bone marrow-derived cells [124,125]. Due to the ability of self-renewal and differentiation into a vast number of cells, CSCs have the property of tumorigenesis induction [126,127]. Gastric CSCs are primarily generated by *CagA*-positive *H. pylori* strain infection [128,129]. High expression of CD44, a compelling marker of CSCs, predisposes cells to the induction of mesenchymal phenotype and EMT [130,131,132,133,134,135,136].

Bessède et al. (2014) studied the role of *H. pylori* in the generation of cells with CSC properties, including several gastric epithelial cell lines (AGS, MKN-45, MKN-74) [103]. The role of *CagA* in the induction of CSC properties was studied on *H. pylori* 7.13 wild-type (WT) *CagA*-positive strain and its knock-out mutants—7.13*CagA*-negative and 7.13D*CagE*-negative *H. pylori* strains. Only the wild-type (WT) (7.13*CagA*-positive) strain-induced mesenchymal changes and EMT in cells; cells infected by *CagA*-positive strains, presented higher expression of mesenchymal markers—CD44, Snail1, vimentin, or zinc finger E-box binding homeobox 1 (ZEB1), while an expression of epithelial markers—cytokeratin 7 (CK7) or osteopontin (SSP-1) was lowered [103]. The migration and invasion properties of cells were enhanced during infection by *CagA*-positive *H. pylori* strains. The relevance of the infection by *CagA*-positive *H. pylori* strain on EMT in gastric cancer was confirmed by significantly higher expression of CD44 and mesenchymal markers in tumor samples.

The Wnt/β-catenin signaling pathway is involved in *CagA*-positive *H. pylori* EMT and the induction of CSC properties [137]. Likewise, chronic *CagA*-positive *H. pylori* infection with N-nitrosoguanidine (MNNG) stimulation causes CSC features with dysplastic lesions and mesenchymal phenotype. Amieva and Peek (2016) observed that CD44+ cells with CSC features displayed increased *CagA* half-life [26]. Accumulation of intracellular *CagA* was confirmed in CD44+ gastric CSCs [138]. *H. pylori*-infected (*CagA*-positive) gastric cancer cells exhibit CSC properties via increased expression of surface markers—CD44 and *Lgr5* with Nanog, Oct4, and *c-myc* upregulation [139,140]. *CagA* impairs transcription factor CDX1 expression, promoting EMT, and CSC features in gastric epithelial cells [141].

### 2.3. CagA and Yes-Associated Protein Pathway

Yes-Associated-Protein (YAP) is an element of the Hippo tumor suppressor signaling pathway, which plays a crucial role in the maintenance of the proper size of organs, tissue homeostasis, cell proliferation, and stem cell maintenance [142,143,144]. Since YAP induces the progression of carcinogenesis, it is considered to be an oncogenic protein [145,146,147]. Increased expression of YAP activates several oncoproteins including connective tissue growth factor (CTGF), cysteine-rich angiogenic inducer 61 (CYR61), or *MYC* oncogene [148,149]. Besides, the overexpression of YAP lowers the expression of epithelial markers including E-cadherin, which induce mesenchymal changes and EMT [150].

Li et al. (2018) showed a significantly higher expression of YAP and TAZ (YAP paralog) in cancerous gastric tissues with a positive correlation between YAP, TAZ levels, and tumor size [151]. During early tumor stages, YAP is expressed mainly in the cell cytoplasm, whereas in advanced stages it is expressed in the nucleus. *CagA*-positive *H. pylori* strain induces higher YAP expression and decreases E-cadherin, N-cadherin, and Slug levels, promoting EMT.

Other studies have shown a potential role of GSK3/β-catenin or AMP-activated protein kinase (AMPK) pathways, which are regulated by *CagA*, in the enhancement of the YAP pathway [152,153]. Molina-Castro et al. (2020) showed that infection by *CagA*-positive *H. pylori* upregulates *YAP1* and large tumor suppressor 2 (*LATS2*) expression in gastric epithelial cells [154]. Overexpression of the oncogenic *YAP1* is associated with aggressiveness and poor prognosis [155]. Gastric cancer development and progression are further promoted by the dysregulated *YAP1*/SLC35B4 axis; another mechanism involves altered IL-1β levels [156,157]. These findings show that YAP might become a potential target in gastric cancer treatment.

## 3. Tumor Necrosis Factor *α*-Inducing Protein of *H. pylori* in EMT

*H. pylori* strains produce a high quantity of tumor necrosis factor-*α* (TNF-*α*)-inducing protein (Tip*α* protein), which binds to the cell surface nucleolin and induces carcinogenic alterations [158,159,160,161,162,163]. It makes up one of the potential markers of *H. pylori* virulence [164,165]. By combining with the nucleolin receptor, Tip*α* induces mesenchymal changes of cells via EMT progression [166,167,168]. Tip*α*, as a carcinogenic factor, activates NF-κB, promoting gastric carcinogenesis by inhibiting miR-3178 expression [169,170]. Besides TNF-*α* release, Tip*α* also induces the expression of several chemokines, including *chemokine* (C-C motif) ligand 2 (Ccl2), *chemokine* (C-C motif) ligand 7(Ccl7), *chemokine* (C-C motif) ligand 20 (Ccl20), C-X-C motif *chemokine* 11 (Cxc11), *chemokine* (C-X-C motif) ligand 2 (Cxcl2), C-X-C motif *chemokine* 5 (Cxcl5), and C-X-C motif *chemokine* 10 (Cxcl10) [171,172,173,174]. EMT is promoted by TGF-β, hepatocyte growth factor (HGF), tumor necrosis factor-*α* (TNF-*α*), or hypoxia-inducible factor 1*α* (HIF1*α*) [175,176]. In gastric carcinogenesis, Tip*α*, along with *CagA* and VacA, plays an underlying role primarily in mucosal damage [177,178].

Suganuma et al. showed that *H. pylori* infection increases the quantity of Tip*α* protein in gastric cancer tissue, inducing tumor progression in *H. pylori* carcinogenesis [161]. Tip*α* reduces cell stiffness and phosphorylates various oncoproteins; it also enhances filopodia formation, morphological, and conformational changes within cells, and expression of vimentin via MEK-ERK phosphorylation, confirming its role in EMT progression [179,180].

Researchers have suggested the role of Tip*α* and nucleolin receptor (88 kDa) on the migration properties of gastric cancer cells [161,181]. Fujiki et al. showed that EMT phenotype induced by Tip*α* could be inhibited via small interfering RNAs targeted for the nucleolin receptor, showing its role in EMT progression [182]. Thus, potential ligands of nucleolin receptors (e.g., lactoferrin) might inhibit *H. pylori* infection by inhibiting *Tipα* and nucleolin interaction [183,184].

Tip*α* induces the formation of filopodia and a reduction of cell stiffness, predisposing cells to higher motility [185]. It has been reported that Tip*α* induces higher levels of IL-1β, TNF-*α*, and IL-8 in SGC7901 cells [164]. Furthermore, *H. pylori* infection promotes the Il-6/STAT3 signaling pathway in AGS cells, which is one factor promoting EMT [186]. Inoue et al. observed that vaccinations with Tip*α* did successfully eradicate *H. pylori* infection [187].

### Antigenic Lpp20 Protein

The Lpp20 protein plays an important role as an *H. pylori* virulence factor, promoting its colonization and survival properties [188]. Lpp20 is a lipoprotein embedded in the external membrane of *H. pylori*, which enhances the stimulation of cell proliferation. Lpp20 expression is increased in the acidic pH of stomach juice and during iron depletion [189,190].

Lpp20 (a structural homologue of Tip*α*), is one of the carcinogenic factors released by *H. pylori* via NF-κB pathway activation [183]. Tip*α* and Lpp20 induce EMT mainly by stimulating cell proliferation, migratory properties of cells, and filopodia formation [191].

Even though Lpp20 and Tip*α* have different locations within the bacterium (external and inner membranes respectively), these proteins show similar effects on gastric cancer cells by stimulating EMT and gastric cancer progression.

## 4. Afadin Protein Downregulation

Afadin is a cytoplasmic actin-filament-binding protein responsible for the formation and stabilization of tight and adherens junctions [192,193,194,195]. Marques et al. (2018) observed that infection by *H. pylori* lowered the expression of Afadin protein independently of *CagA* and VacA in MKN74 and NCI-N87 cell lines [196]. Downregulation of Afadin resulted in dysregulation of tight junctions and adherens junctions, which caused inappropriate cell permeability, stiffness, and impaired transepithelial electrical resistance. The decreased level of Afadin induced higher expression of Snail (EMT marker) and increased actin stress fiber formation; no changes were observed in Slug, ZEB1, vimentin, and N-cadherin levels. So far, this is the first study suggesting a role of Afadin in the EMT induction in gastric cancer cells infected by *H. pylori*.

## 5. Penicillin-Binding Protein 1A

Penicillin-binding protein (PBP) is a specialized acyl serine transferase released by *H. pylori* with an affinity for penicillin binding; any modifications within PBP might induce resistance to β-lactam antibiotics (amoxicillin, penicillin G, ampicillin) [197]. These mainly include either mutations or mosaics (or both) of *PBP2X*, *PBP2B*, and *PBP1A* [198]. It was shown that *H. pylori* PBPs might present unique characteristics because of several putative genes encoding PBPs.

Huang et al. (2019) showed significant differences between *PBP1A* mutation-positive *H. pylori* (*H. pylori CagA*+/P+) strain and *CagA*+/P− in terms of clinicopathological characteristics of gastric cancer and EMT induction [199]. *H. pylori CagA*+/P+ infection resulted in progressive and clinically significant EMT and gastric cancer severity. Researchers have observed decreased levels of E-cadherin and increased α-smooth muscle actin (*α*-SMA) levels; likewise, it was observed that *PBP1A* mutation is associated with decreased miR-134 levels. MiR-134 suppresses gastric carcinogenesis by targeting the *KRAS* gene and Golgi phosphoprotein 3 (GOLPH3) downregulation; another essential target gene of miR-134 is *FoxM1* [200,201]. The inhibition of PBP via paepalantine is suggested to be a promising method of *H. pylori* activity decline [202]. Marcus et al. showed that at pH 3.0, the expression of PBPs genes is significantly decreased [203].

## 6. Upregulation of MicroRNA-29a-3p

Micro-RNAs constitute a family of small noncoding RNAs, which are regulators of posttranscriptional gene expression by binding to the 3′-untranslated region of mRNA, promoting an inappropriate translation of targeted mRNA. There is an increasing interest in micro-RNA involvement in many diseases and malignancies [204,205,206,207]. A vast number of micro-RNAs are directly associated with the progression of gastric cancer, including microRNA-4513, microRNA-95, and microRNA-4268 [208,209,210].

Sun et al. presented the relevance of upregulation of microRNA-29a-3p and its role in the decreased expression of A20 in patients infected by *H. pylori* [211]. A20 is the ubiquitin-editing enzyme, which negatively regulates the NF-kB pathway, controlling proper cellular activation, development, and differentiation [212]. *H. pylori* infection induced the overexpression of microRNA-29a-3p in gastric cancer tissue samples, which resulted in an enhanced ability of the migration of gastric epithelial cells. Furthermore, the silencing of A20 increased the EMT markers Snail, vimentin, and N-cadherin, and decreased E-cadherin levels. The authors have suggested that the overexpression of micro-RNA-29a-3p might be associated with a higher probability of EMT induction in gastric cells.

## 7. Downregulation of Programmed Cell Death Protein 4

Programmed cell death protein 4 (PDCD4) is a tumor suppressor responsible for the inhibition of cell growth, tumor invasion, metastasis, or induction of apoptosis [213]. It is localized in the nucleus of proliferating cells and binds to the eukaryotic initiation factor-4A (eIF4A) or eukaryotic translation initiation factor 4 G (eIF4G), inhibiting translation [214,215]. PDCD4 controls the inhibition of cancer invasion by regulating the expression of mitogen-activated protein 4 kinases 1 or urokinase plasminogen activator receptor [216]. It is also responsible for the inhibition of the Y-box binding protein expression and connecting with the DNA-binding domain of the *TWIST1* gene [217].

PDCD4 is one factor responsible for the induction of carcinogenesis [218]. Yu et al. showed that the downregulation of PDCD4 expression resulted in a decreased level of E-cadherin, increased *TWIST1*, and vimentin levels in gastric cancerous tissues [219]. PDCD4 downregulation and subsequent alterations in epithelial–mesenchymal markers were induced by *CagA*-positive *H. pylori* strain infection. PDCD4 overexpression reduced EMT. MicroRNA-21 overexpression in gastric cancer modulates the expression of the tumor suppressors phosphatase and tensin homolog (*PTEN*) and PDCD4, altering molecular pathways associated with cell growth, invasion, migration, and apoptosis [220].

## 8. Upregulation of lysosomal-Associated Protein Transmembrane 4β

Lysosomal-associated protein transmembrane 4β (*LAPTM4B*) is a proto-oncogene relevant in the progression of tumorigenesis, regulating molecular mechanisms, and cellular functioning, including proliferation, migration, and invasion [221]. *LAPTM4B* upregulation has been observed in various cancers, including hepatocellular carcinoma, breast cancer, and lung adenocarcinoma [222,223,224].

It has been reported that *H. pylori* infection leads to the aberrant expression of *LAPTM4B* in gastric epithelial cells, which eventually induces the EMT process and further malignant transformation [225]. Researchers have observed that *LAPTM4B* upregulation by *H. pylori* infection increases the levels of EMT markers (N-cadherin, vimentin) while lowering E-cadherin levels and dysregulating cell–cell junctions. Additionally, there was a noticeable increase of ZEB1, β-catenin, Snail, and Slug markers, whereas tight junction protein-1 (zonula occludens-1, ZO-1) level was decreased. *LAPTM4B* seems to be a factor strictly associated with EMT induction since it is responsible for regulating cells’ filopodia, increasing motility of gastric cancer cells, and a higher invasion potential [226]. Overexpression of *LAPTM4B* results in poorer prognosis in various cancers [227].

## 9. Cancer-Associated Fibroblasts

Physiologically, natural gastric stroma contains only a few fibroblasts, and those are mainly myofibroblasts. The number of fibroblasts is significantly increased in tissues affected by inflammation or neoplasm [228,229,230,231]. In a gastric cancer microenvironment, macrophages via interaction with mesenchymal stem cells induce differentiation of cancer-associated fibroblasts (CAFs) [232,233,234]. CAFs are crucial in the initiation, growth, and migration properties of gastric cancer cells [235,236]. They are responsible for the release of carcinogenic and proinflammatory factors including interleukin-6 (Il-6), cyclooxygenase-2 (COX-2), chemokine (C-X-C motif) ligand 1 (Cxcl1), chemokine (C-X-C motif) ligand 9 (Cxcl9), interferon gamma-induced protein 10 (Cxcl10), stromal cell-derived factor 1 (Cxcl12), and fibroblast-specific protein 1 (FSP1), which promote the EMT process and induce carcinogenesis, promoting migration and invasion [237,238,239]. Activation of death receptor 4 (TRAIL receptor 1, TRAILR1) and ensuing activation of caspase 8 by CAFs induce apoptosis in gastric cells [240,241]. CAFs also facilitate angiogenesis via the production of proangiogenic factors-fibroblast growth factor (FGF), vascular endothelial growth factor (VEGF), interleukin-8 (IL-8), and stromal cell-derived factor 1 (SDF-1) [242,243,244]. Angiogenesis is enhanced by the upregulation of HIF-1*α* in fibroblasts, which is common during *CagA*+VacA+ *H. pylori* infection [245].

*H. pylori* infection results in the transformation of fibroblasts and myofibroblasts into CAFs [233]. *CagA*+VacA+ *H. pylori* infection results in the overexpression of fibroblast activation protein (FAP), fibroblast surface protein (FSP) mRNA, and increased levels of the proinflammatory factors IL-6, IL-8, COX-2, and SDF-1 [246]. Overexpression of FAP alters the regulation of fibroblast growth, impairs tissue repair, induces EMT progression of gastric cancer cells, and epithelial carcinogenesis [247,248,249]. Krzysiek-Maczka et al. observed the overexpression of mesenchymal markers (*α*-SMA, N-cadherin, vimentin, Snail, Twist) and lowered levels of E-cadherin, antiapoptotic B-cell lymphoma 2 (Bcl-2), and proliferative marker Ki-67 protein within fibroblasts [246]. *CagA*+VacA+ *H. pylori* infection resulted in the aberrant apoptosis process, an increased level of collagen production, and impaired fibroblast and myofibroblast differentiation into CAFs. This induced desmoplastic reactions and EMT within gastric cells, promoting carcinogenesis. The increased levels of β1-integrin and COX-2, which are crucial in invasion, metastasis, and angiogenesis, enhanced the migration properties of the infected cells. *CagA*+Vac+ *H. pylori* infection results in the upregulation of HIF-1*α* and tenascin-C (TNC), enhancing EMT and tumor progression, the release of proangiogenic factors, and further CAFs activation [250,251,252]. *CagA*+Vac+ *H. pylori* infection activates gastric fibroblasts and induces the release of TGF-β [253].

The tumor-promoting ability of CAFs is enhanced by microRNA dysregulation [254]. CAFs stimulate EMT in gastric cancer cells in a microRNA-214-dependent manner—it induces tumor-promoting ability of CAFs through fibroblast growth factor 9 (FGF9) targeting [255,256]. Likewise, suppression of microRNA-149 (through regulation of IL-6 levels) induces EMT, promoting protumor activity of CAFs [257]. CAFs also promote EMT and gastric carcinogenesis via activation of erythropoietin-producing hepatocellular A2 receptor (EphA2) and IL-6-JAK2-STAT3 signaling pathways [258,259]. CAFs release a significant amount of galectin-1, which promotes EMT and gastric cancer progression by binding to β1 integrin [260]. EMT is also promoted by CAFs-derived IL-33 through the activation of the ERK1/2-SP1-ZEB2 pathway [261].

## 10. Heparin-Binding Epidermal Growth Factor and Matrix Metalloproteinase-7

Heparin-binding epidermal growth factor (HB-EGF) is an epidermal growth factor receptor (EGFR) cognate ligand with mitogenic and chemotactic properties for fibroblasts and smooth muscle cells [262]. *H. pylori* infection results in the upregulation of HB-EGF gene expression, which is a potential serological biomarker for gastric cancer [263,264,265,266,267]. HB-EGF increases cell motility, invasiveness, and metastatic properties of gastric cells [268]. HB-EGF upregulation is because of the activation of *H. pylori* g-glutamyltranspeptidase, activating PI3K, and p38 kinase-dependent signal transduction pathways [269].

Matrix metalloproteinase-7 (MMP-7) is an enzyme responsible for the degradation of adherence between cells in the extracellular matrix, and overexpression of MMP-7 induces a more aggressive course of gastric cancer [270,271,272,273]. MMP-7 and HB-EGF expression is upregulated by *H. pylori* infection and is associated with EMT [274,275]. Likewise, MMP-7 overexpression is because of increased levels of gastrin, activator protein 1 (AP-1), and NF-kB induced by *H. pylori* infection [276,277,278,279]. Levels of MMP-7 might correlate with the degree of rehabilitation after treatment and poor prognosis in gastric cancer survival [270,280].

Yin et al. showed that excessive gastrin secretion due to *H. pylori* infection results in the increased release of MMP-7 and soluble HB-EGF, promoting EMT [281]. HB-EGF might promote tumor growth and EMT by IL-4-conditioned media from macrophages [282]. Besides HB-EGF, other ErbB ligands (transforming growth factor-α (TGF-α) and amphiregulin (AREG)) have a prognostic impact on gastric cancer and might make up potential targets in cancer therapy [283,284,285]. Furthermore, trastuzumab sensitivity in gastric cancer can be predicted by HB-EGF expression [286]. Gastric cancer invasion can be increased by nuclear translocation of the cytoplasmic domain of HB-EGF [287,288]. However, Krakowiak et al. showed that MMP-7 might limit *H. pylori*-induced inflammation and damage via M1 macrophage polarization suppression [289].

## 11. Stem Cells

Mesenchymal stem cells (MSCs) present a multipotent potential to differentiate into various cell types, while keeping a capacity for self-renewal [290,291]. MSCs are recruited to the areas of *H. pylori* infection, neoplasia, or cancer, being involved in the migration properties of gastric cancer cells [292,293,294]. MSCs promote EMT, tumor growth, and metastasis [295,296]. Nanog Homeobox Retrogene P8 (NANOGP8) makes up the main regulator of gastric CSCs [297].

Zhang et al. showed that human umbilical cord MSCs (hucMSCs) infected by *H. pylori* gained a fibroblastic phenotype, enhancing EMT progression in gastric cancer cells [298]. Furthermore, hucMSCs infected by *H. pylori* showed a significant decrease in the levels of E-cadherin and overexpression of mesenchymal markers (N-cadherin, vimentin) and inflammatory cytokines (IL-8, IL-6, IL-1β, TNF-*α*, platelet-derived growth factor subunit B (PDGF-B), VEGF, epidermal growth protein (EGF), granulocyte-macrophage colony-stimulating factor (GM-CSF), and monocyte chemoattractant protein 1 (MCP-1). High concentrations of IL-15 secreted by gastric cancer MSCs contribute to tumor cell metastasis; increased IL-17 levels promote gastric cancer invasiveness [299,300].

Increased expression of *SOX4* induces EMT via a release of TGF-β and contributes to the development of stem cell characteristics of gastric cancer cells [301]. Furthermore, CSCs, which display mesenchymal phenotypes, overexpress EMT markers such as Twist, Snail, or ZEB [302]. *CagA*-positive *H. pylori* infection induces EMT, and CSCs feature in gastric mucosal cells via overexpression of the Wnt/β-catenin signaling pathway [137]. The progression of CSCs is stimulated by EMT-related CD44+ cells’ emergence [303,304,305,306,307]. It was hypothesized that overexpression of the regulator of G-protein signaling pathway 1 (RGS1) might promote the transition of stem cells into CSCs [308]. Besides, CSCs are involved in EMT progression via a vast number of mechanisms, including an overactivation of *KRAS*, *STAT3*, *Rac1*, *Wnt*, *Notch*, *PTEN*, ERK, NF-κB signaling pathways, or hypoxic microenvironment [309,310,311,312,313,314,315,316,317,318]. Recently, researchers have proposed the role of CSC-associated protein, leucine-rich repeat and immunoglobulin-like domain-containing nogo receptor-interacting protein 2 (LINGO2), in gastric cancer and EMT initiation [319].

## 12. Conclusions

The association between *H. pylori* infection and gastric carcinogenesis remains controversial. In fact, some mechanisms are yet not fully understood. *H. pylori* infection seems to play an important role in terms of EMT induction within gastric mucosa (Figure 2).

The severity and progression of gastric cancer depends on the presence of specific *H. pylori* virulence factors. Likewise, cellular components that are associated with EMT progression can be influenced by *H. pylori* infection (Table 1). EMT and its progression depend on the particular strain of *H. pylori* along with its properties and virulence factors (Table 1). Specific strains may promote EMT with various intensity, thus influencing the eventual outcome of gastric cancer differently. Even though most known virulence factors, along with cellular components and their mechanisms of action, have been studied, more research should be performed to investigate the relationships between those factors, as well as to search for unknown processes and aspects. Notwithstanding, there is potential for further pharmacological research since all virulence factors constitute potential molecular targets for new medications. This may eventually provide new treatment strategies for gastric cancer, along with the possible eradication of *H. pylori* during the early stages of infection, which could significantly decrease the number of gastric cancer incidents.

## Figures and Tables

**Figure 1 cells-09-01055-f001:**
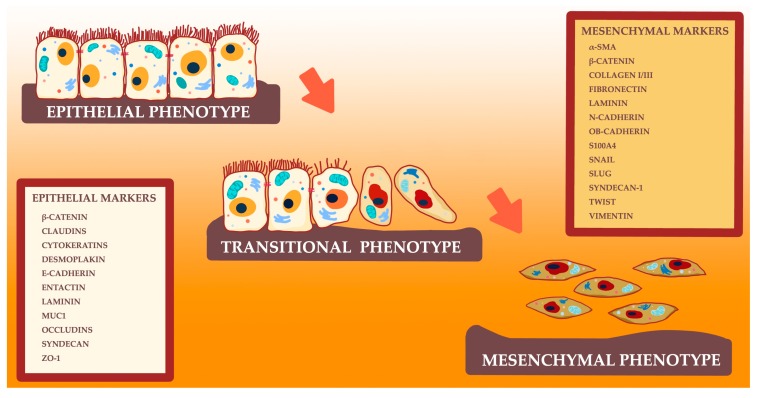
Schematic of the epithelial–mesenchymal transition and chosen epithelial and mesenchymal markers.

**Figure 2 cells-09-01055-f002:**
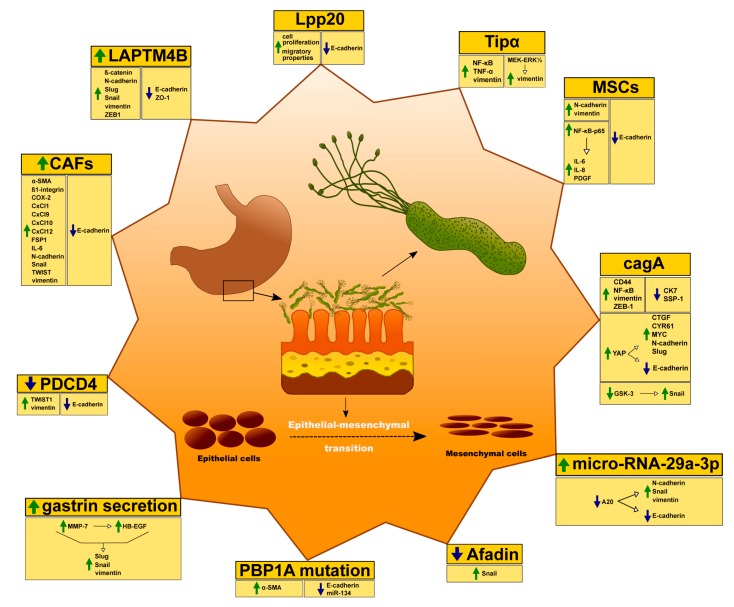
Possible mechanisms of the association between *H. pylori* infection and EMT.

**Table 1 cells-09-01055-t001:** Virulence factors and cellular components associated with EMT and *H. pylori* infection.

Factors	Increase	Decrease
Afadin	actin stress fibers; Snail	ND
CAFs	*α*-SMA; Collagen I; Collagen III; COX-2; FAP; FGF-2; FSP1; HGF; HIF-1α; IL-6; IL-8; Integrin β-1; N-cadherin; SDF-1; Snail; TGF-β; TNC; *TWIST*; VEGF; vimentin	Bcl-2; Ki67
*CagA*	CD44; Snail 1; vimentin; ZEB1	CK7; SSP-1
*CagA* & YAP pathway	CTGF; CYR61; *MYC*; N-cadherin; Slug;	E-cadherin
HB-EGF & MMP-7	AP-1; NF-κB; Slug; Snail; vimentin	ND
*LAPTM4B*	β-catenin; N-cadherin; Slug; Snail; vimentin; ZEB1	E-cadherin; ZO-1
Lpp20	ND	E-cadherin
miR-29a-3p	N-cadherin; Snail; vimentin	A20 gene; E-cadherin
MSCs	EGF; GM-CSF; IL-1β; IL-6; IL-8; MCP-1; N-cadherin; PDGF-B; TNF-*α*; VEGF; vimentin	E-cadherin
*PBP1A* mutation	*α*-SMA; *FoxM1*	E-cadherin; miR-134
PDCD4	Twist1; vimentin	E-cadherin
Tip*α*	Ccl2; Ccl7; Ccl20; Cxcl1; Cxcl2; Cxcl5; Cxcl10; IL-1β; IL-6/STAT3 pathway; IL-8; N-cadherin; NF-κB; TNF-*α*; vimentin	E-cadherin

ND—no data.

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
