# Peer review of "Mechanisms of the Epithelial–Mesenchymal Transition and Tumor Microenvironment in *Helicobacter pylori*-Induced Gastric Cancer"

_cells, 2020, doi:10.3390/cells9041055_

Round 1
Reviewer 1 Report
no other comments
Author Response
Jacek Baj
Medical University of Lublin
Department of Human Anatomy
Jaczewskiego 4
20-090 Lublin, Poland
jacek.baj@umlub.pl
16th April 2020
Dear Reviewer,
Thank you very much for reviewing our manuscript. We appreciate the interest and commitment you have provided for this work.
We are pleased to submit the manuscript entitled ‘Mechanisms of the epithelial-mesenchymal transition in Helicobacter pylori-induced gastric cancer’.
We have added one new figure and the manuscript has been checked once again in terms of English by one of our native speaker colleagues.
We hope that after this revision, the manuscript is of a higher quality and worth reading.
We wish you all the best!
Sincerely,
Jacek Baj
on behalf of the authors
Reviewer 2 Report
This review comprehensively summarized the mechanisms of the epithelial-mesenchymal transition and tumor microenvironment in Helicobacter pylori-induced gastric cancer, although some mechanisms were non specific for H pylori. One question was that figure 2 was presented in the text, figure 1 was missing. Is it a typo?
Author Response
Jacek Baj
Medical University of Lublin
Department of Human Anatomy
Jaczewskiego 4
20-090 Lublin, Poland
jacek.baj@umlub.pl
16th April 2020
Dear Reviewer,
Thank you very much for reviewing our manuscript. We are very grateful for the commitment you have provided during the revision of this manuscript.
We are pleased to submit the manuscript entitled ‘Mechanisms of the epithelial-mesenchymal transition in Helicobacter pylori-induced gastric cancer’.
The manuscript has been checked once again in terms of English by one of our native speaker colleagues and we hope that now it is much more improved. We have also added the missing figure. We have added information in the introduction that the described mechanisms are not only specific to H. pylori – we hope that it will clarify it for the readers.
We hope that after this revision, the manuscript is of a higher quality and worth reading.
We wish you all the best!
Sincerely,
Jacek Baj
on behalf of the authors
This manuscript is a resubmission of an earlier submission. The following is a list of the peer review reports and author responses from that submission.
Round 1
Reviewer 1 Report
Dear Editor,
the review of Baj et al. reporting very interesting data about the relationship between HP infection and epithelial to mesenchymal transition phenomenon in gastric cancer.
Literature analysis is innovative involving the most recent papers on this theme.
The review is well written and, in my opinion, suitable for the publication in Cells after the following minor revision:
Add more information about the link among HP infection, immune response and EMT A figure showing possible mechanisms of the association between HP infection and EMT could improve the manuscript A short overview about the possible role of EMT in human carcinogenesis could improve the appeal of the manuscript (PMID: 31718020 - PMID: 26563370 - PMID: 31832068 - PMID: 31786880 - PMID: 31614568).
Author Response
Jacek Baj
Medical University of Lublin
Department of Human Anatomy
Jaczewskiego 4
20-090 Lublin, Poland
jacek.baj@umlub.pl
23rd December 2019
Dear Reviewer,
Thank you very much for reviewing our manuscript. We appreciate the interest and commitment you have provided for this work. We are very grateful for your extremely precious comments. We are convinced that thanks to your suggestions this manuscript will be much more valuable.
We are pleased to submit explanations and details of our revisions in the manuscript entitled ‘Mechanisms of the epithelial-mesenchymal transition in Helicobacter pylori-induced gastric cancer’.
The followings are our point-by-point responses:
We have added more information about the link between H. pylori infection, immune responses and further induction of EMT in lines 88-94. We have also added several references [40 - 44] with regard to this suggestion.
Further, we have added a new figure (Figure 2), originally made by the authors, therefore, no copyrights are needed. We have entitled this figure ‘Several possible mechanisms of the association between H. pylori infection and EMT’, as you have suggested.
A short overview of the role of EMT in human carcinogenesis was added in the following lines: 76-87. We have also added several new references [25 - 39].
We would like to thank you again for your effort, feedback and extremely helpful comments.
We wish you all the best and Happy New Year!
Sincerely,
Jacek Baj, MD, PhD
on behalf of the authors
Reviewer 2 Report
Interesting revision of the literature about factors involved in epiyhelial-mesenchymal transition during H. pylori infection including focus on some HP virulence factors. Some comments must be due, expecially regarding the introduction and conclusion sections.
Introduction: Please clarify the phrase from line 50 to 52: diseases that are not associated with the presence of HP are also cyted (i.e. esophageal cancer), hence the concept of "responsibility"should be modulated in order to include all the cyted pathologies, or, better, should be addressed only to pathologies in which HP intervenes with direct or indirect mechanisms.
Please, document the association between HP and vesicular lithiasis or duodenitis.
Lines 62-65: please document the statements with appropriate references. What do you mean with the expression"in spite of appearance"? It doesn't sound scientific.
Lines 63-66: The object must ground on clear literature: there is an intrinsic contraddiction between the phrase that claims the existence of a significant number of HP virulence factors associated with EMT promotion in gastric cells and the phrase that states the absence of literature concerning the role of HP on EMT. In this way it seems that the object does not rest on a clear basis.
Lines 91-105: please, specify the literature that document the action of CagA- and CagE- strains: are they natural strains or are the terms referring to the previously mentioned knock-out strains?
Conclusions: The comments and table 1 should be better contextualized along the paragraph: indeed, the review and the table include not only HP virulence factors but also cellular components associated to EMT and that can be influenced by HP.
Please check the form and propriety of the references (for example, refs 10 and 11 are not appropriate in the context).
Author Response
Jacek Baj
Medical University of Lublin
Department of Human Anatomy
Jaczewskiego 4
20-090 Lublin, Poland
jacek.baj@umlub.pl
23rd December 2019
Dear Reviewer,
Thank you very much for reviewing our manuscript. We are very grateful for the interest and commitment you have provided during the revision of this manuscript. We appreciate your precious comments as well as constructive suggestions.
We are pleased to submit explanations and details of our revisions in the manuscript entitled ‘Mechanisms of the epithelial-mesenchymal transition in Helicobacter pylori-induced gastric cancer’.
The followings are our point-by-point responses:
Ad. Introduction:
The phrase from line 50 to 52: only pathologies associated with H. pylori infection were left in this phrase. Esophageal cancer, vesicular lithiasis and duodenitis were removed from this phrase because we checked again the cited article and it turned out that it was our misunderstanding. Thus, we removed this citation from this phrase and added new ones - [11-14].
Ad. Lines 62-65:
The references [23,24] were added as you have suggested. We also have removed the sentence ‘Yet, in spite of appearances and recent research, there are no many reports regarding the influence of H. pylori on EMT and gastric cancer’.
Ad. Lines 63-66:
We have decided to remove the sentence ‘Yet, in spite of appearances and recent research, there are no many reports regarding the influence of H. pylori on EMT and gastric cancer’ because it does not sound scientific and made a contradiction with the previous sentence. Originally, we have added this sentence, because even though it seems like there is a lot of factors associated with H. pylori infection and EMT, the number of literature and studies is still scarce.
Ad. Lines 91-105:
We have added information about the researcher and the year of the following study (reference [54]). We have also specified the description of the H. pylori strains (lines 115-117).
Ad. Conclusions:
Throughout the manuscript, we have tried to point out that not only H. pylori virulence factors, but also cellular components associated with EMT can be influenced by H. pylori infection. We have changed the order of factors in the table (Table 1) and put in in alphabetical order.
Ad. References:
We have changed the previous [10] reference into the proper citation style. We have used this citation here since in the cited article there was information we have used in our manuscript. We have deleted previous [11] reference and replaced it with proper ones (new references [11-14]).
We would like to thank you again for your effort, feedback and extremely helpful comments.
We wish you all the best and Happy New Year!
Sincerely,
Jacek Baj, MD, PhD
on behalf of the authors
Reviewer 3 Report
The authors have thoroughly summarized the mechanisms of EMT in H pylori-induced gastric cancer, in particular, HP virulence factors cause EMT in cancer progression of gastric stroma. However, there are some concerns:
(1) The association between HP infection and gastric cancer is controversial, and the underlying mechanisms are unclear.
(2) It is known that HP causes gastric mucosa epithelial cell polarization changes and malignant transformation, not as the authors claimed as cancer progression of gastric stroma.
(3) The listed factors associated with EMT in other cancers are barely associated with HP infection and with HP-associated gastric malignant transition.
(4) The virulence factors listed in this manuscript were not well organized.
Author Response
Jacek Baj
Medical University of Lublin
Department of Human Anatomy
Jaczewskiego 4
20-090 Lublin, Poland
jacek.baj@umlub.pl
23rd December 2019
Dear Reviewer,
Thank you very much for reviewing our manuscript. We are very grateful for commitment you have provided during the revision of this manuscript. We appreciate your precious comments as well as constructive criticism.
We are pleased to submit explanations and details of our revisions in the manuscript entitled ‘Mechanisms of the epithelial-mesenchymal transition in Helicobacter pylori-induced gastric cancer’.
The followings are our point-by-point responses:
Ad. 1.
We have emphasized that the association between H. pylori infection and gastric cancer is controversial and still mechanisms are unclear, so in lines: 460-461 we have added this information.
Ad. 2.
We have removed the phrase ‘cancer progression of gastric stroma’ from our manuscript and added a more precise description with the use of the ‘malignant transformation’ phrase.
Ad. 3.
In fact, some of the factors associated with EMT in other cancers that we have mentioned are barely related to H. pylori infection and further gastric carcinogenesis. However, we have only mentioned other cancers just for example. The aim of our work was not to describe those factors in detail, but just to suggest that they might be present.
Ad. 4.
We have changed the order of factors in the table (Table 1.) and put them in alphabetical order. Because all of the paragraphs concern H. pylori infection and possible induction of EMT with further carcinogenesis, and these paragraphs are not strictly associated with one another, so we decided to describe the factors in the following order:
CagA & cancer stem-cell properties CagA & YAP pathway TNF-α-inducing protein Afadin PBP1A & microRNA-134 MicroRNA-29a-3p PDCD4 LAPTM4B CAFs HB-EGF & MMP-7 MSCs Lpp20
We would like to know whether we should change the abovementioned order of the paragraphs?
We would like to thank you again for your effort, feedback and extremely helpful comments.
We wish you all the best and Happy New Year!
Sincerely,
Jacek Baj, MD, PhD
on behalf of the authors